# Test-Time Prompt Tuning for Zero-Shot Generalization in Vision-Language Models

**Manli Shu**[1*]   **Weili Nie**[2]   **De-An Huang**[2]   **Zhiding Yu** [2]
**Tom Goldstein** [1]   **Anima Anandkumar**[2,3,†]   **Chaowei Xiao**[2,4†]
[1] University of Maryland, [2] NVIDIA, [3] Caltech, [4] Arizona State University

## Abstract

Pre-trained vision-language models (e.g., CLIP) have shown promising zero-shot generalization in many downstream tasks with properly designed text prompts. Instead of relying on hand-engineered prompts, recent works learn prompts using the training data from downstream tasks. While effective, training on domain-specific data reduces a model's generalization capability to unseen new domains. In this work, we propose test-time prompt tuning (TPT), a method that can learn adaptive prompts on the fly with a single test sample. For image classification, TPT optimizes the prompt by minimizing the entropy with confidence selection so that the model has consistent predictions across different augmented views of each test sample. In evaluating generalization to natural distribution shifts, TPT improves the zero-shot top-1 accuracy of CLIP by 3.6% on average, surpassing previous prompt tuning approaches that require additional task-specific training data. In evaluating cross-dataset generalization with unseen categories, TPT performs on par with the state-of-the-art approaches that use additional training data. Project page: https://azshue.github.io/TPT/.

## 1   Introduction

Recent advances in vision-language pre-training, such as CLIP [1] and ALIGN [2], present a promising direction for developing foundation models for vision tasks [3, 4]. These foundation models encode a wide range of visual concepts after training on millions of noisy image-text pairs and can be applied to downstream tasks in a zero-shot manner without task-specific training data [5–11]. This is made possible by appropriately designed instruction prompts. Take image classification in Figure 1 as an example: We can prepend a category name with a prompt "a photo of a" (*e.g.,* "a photo of a dog"). Images can then be classified by using CLIP to measure their alignment with the various class descriptions. Designing such prompts thus plays a crucial role in applying foundation models to downstream tasks in a zero-shot manner. However, such hand-crafted prompts require domain-specific heuristics and may not be optimal.

Recent works address this by proposing *prompt tuning* to directly learn prompts using training data from downstream tasks [12]. We can fine-tune prompts with training data in the same way we fine-tune model parameters since prompt embeddings are part of the model input and are differentiable with respect to the loss function. Such an approach can find better prompts compared to hand-crafted ones, but the learned prompts are limited to the distribution and tasks corresponding to training data and may have limited generalization beyond that. In addition, this approach requires training data with annotations, which can be expensive and is not available for zero-shot tasks.

---

*Work done during an internship at NVIDIA. manlis@cs.umd.edu.

†Equal advising

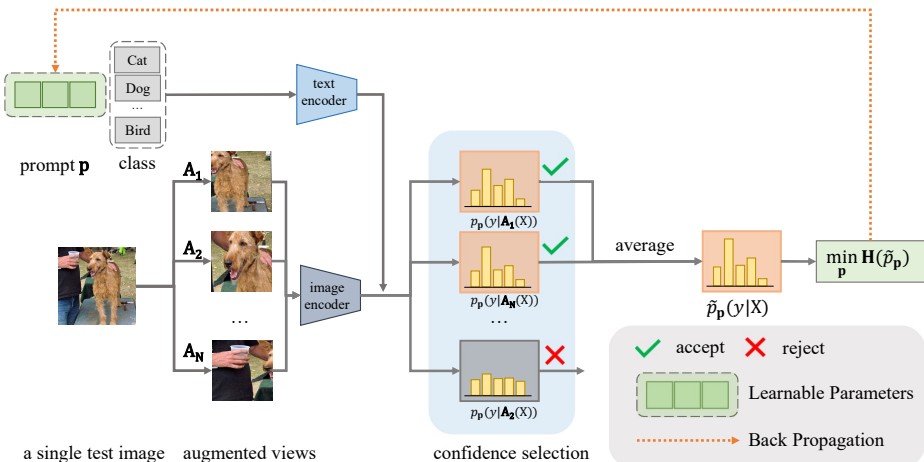

Figure 1: **Test-time Prompt Tuning (TPT)** for image classification. We tune adaptive prompts on the fly with a single test sample, without the need for additional training data or annotations. TPT optimizes the prompt to encourage consistent predictions across augmented views by minimizing the marginal entropy. We introduce *confidence selection* to filter out noisy augmentations.

**Our Approach.** To address the aforementioned challenges, we propose **t**est-time **p**rompt **t**uning (TPT) that tunes the prompt on the fly using only the given test sample. The tuned prompt is adapted to each task, making it suitable for zero-shot generalization without requiring any task-specific training data or annotations. TPT retains the zero-shot generalization setting since no additional training data or annotations are used.

We explore TPT on two different downstream tasks: image classification and context-dependent visual reasoning. For each downstream task, we design a customized test-time tuning strategy that fits the nature of the task. Without loss of generality, we consider CLIP [1] as our vision-language foundation model, for its simplicity in design and its wide applicability [13].

For image classification, a test sample is an input image. Given a single sample at test time, we perform prompt tuning by generating multiple randomly augmented views, and optimizing the text prompt so that the model has consistent predictions across different augmented views. This is done by minimizing the marginal entropy among the outputs of the augmented views. In addition, since some augmentations may lead to misleading model predictions, we propose *confidence selection* to filter out those "noisy" augmented views. We discard augmented views with a high prediction entropy (*i.e.*, low confidence), and only include high confidence views in the consistency optimization.

We evaluate the zero-shot generalization of TPT in two image classification settings: natural distribution shifts [14] and cross-dataset generalization [15]. For the setting of evaluating natural distribution shifts, TPT boosts the Top-1 accuracy of CLIP in the zero-shot setting by 3.6% on average compared to using a hand-crafted prompt, achieving on-par accuracy with previous prompt tuning methods that require additional training data (*i.e.*, ImageNet). TPT achieves a maximum improvement of 6.9% on ImageNet-A compared to using a hand-crafted prompt, surpassing the existing few-shot prompt tuning method by 5.1%. For the setting of evaluating cross-dataset generalization with possibly unseen categories, TPT achieves on-par performance with the state-of-the-art few-shot prompt tuning method [15] without the need for additional training data or annotations.

For the second task of context-dependent visual reasoning, such as Bongard-HOI [16], a test sample contains two sets of support images and a query image for evaluation. The two sets of support images exemplify the presence and the absence of a human-object interaction (HOI) concept (*e.g.,* "ride bike"). The model is then asked to infer whether the query image contains the underlying concept. Given such a test sample, we apply TPT by tuning prompts to better differentiate between the two support sets, so that the query image can be better classified (Figure 4). Despite the use of support sets, our approach is still considered zero-shot for visual reasoning, because we do not use either training tasks from other concepts or the annotation of the query image at test time to update the prompt of the test task. By adapting TPT to context-dependent visual reasoning, we outperform the state-of-the-art method [17] by 4.1% Bongard-HOI benchamrk [16].

We summarize our main contributions as follows:

- We propose test-time prompt tuning (TPT) that does not need any training data or annotations to optimize the prompt. To the best of our knowledge, we are the first to perform prompt tuning on a single test sample in a zero-shot manner.
- We introduce confidence selection as a simple plug-and-play module of TPT for image classification. It improves entropy minimization among augmented views by filtering out "noisy" augmentations that lead to low-confidence predictions.
- We conduct extensive experiments on image classification under natural distribution shift, cross-dataset generalization, and context-dependent visual reasoning. TPT improves CLIP in a zero-shot manner to be on par with prompt tuning methods that require additional training data.

## 2   Related Work

**Prompting for foundation models.**   Foundation models are those trained on large-scale hetero-geneous data, of which the knowledge can be transferred to various downstream tasks in natural language processing [18, 19], computer vision [1, 2, 20], etc. Recent work has proposed different ways to efficiently and effectively transfer such knowledge to downstream task [21–24]. Prompting is a heuristic way to directly apply foundation models to downstream tasks in a zero-shot manner. Prompt works as a part of the text input that instructs the model to perform accordingly on a specific task. However, such zero-shot generalization is highly dependent on a well-designed prompt. Prompt tuning [12, 25–27] proposes to learn the prompt from downstream data in the continual input embed-ding space, which presents a parameter-efficient way of fine-tuning foundation models. Although initially developed for language models, prompt tuning has later been applied to other domains, including vision-language models [28, 15, 29] and continual learning [30]. CoOp[28] applies prompt tuning to CLIP. By tuning the prompt on a collection of training data, CoOp effectively improves CLIP's performance on the corresponding downstream tasks. CoCoOp [15] points out that CoOp lacks in generalization to out-of-distribution data, and proposes to alleviate the problem by making the prompt conditioned on model inputs. Despite being effective on the given task, this line of work requires access to downstream training data with annotations, restricting the zero-shot knowledge transfer of foundation models. Another line of work proposes to tune the prompt in an unsupervised manner [31, 32]. However, it requires access to *multiple* samples from either the training or testing split. In this work, we propose test-time prompt tuning that works on a *single* test sample. Our method can directly work with the zero-shot applications of foundation models.

**Generalization under data distribution shifts.**   A reliable machine learning model is supposed to perform well under data distribution shifts for real-world applications. For a model trained on a given set of data, distribution shift refers to the discrepancy between the underlying distributions of the test and the training data. Distribution shifts can occur naturally in the real world due to variations in the environment [33] or the encounter of unseen concepts [34]. For example, in the meta-learning literature [35], each test sample consists of a novel task (i.e., distribution), and the models should be able to quickly adapt to the novel distributions. Even in the standard evaluation protocol for machine learning models, there exists a subtle difference in the data distribution between the training and testing splits [36, 37], which is also one type of distribution shift. Pre-trained vision-language models like CLIP can generalize to downstream tasks with various distribution shifts in a zero-shot manner. Such zero-shot generalization ability presents a promising direction for realizing reliable and generic machine learning models. Our method aims to improve CLIP towards a better generic model in this work, instead of adapting it to specific downstream tasks or target datasets. We leverage the assumption that a robust model should have decision boundaries lying in low-density data regions [38]. Consistency-regularization-based methods [39, 20] achieve this goal by making the network outputs invariant to small input noises. For classification tasks, we use consistency regularization as our test-time prompt tuning objective with the confidence selection module.

**Test-time optimization.**   The idea of adapting machine learning models to test samples on the fly has been applied to different tasks [40–43]. This work mainly focuses on applying the technique to improve model generalization. One challenge in this area is to design a practical test-time objective. Test-time training and its variants [44, 45] modify the training objective and the network architecture by adding a self-supervised multi-task branch, which will then be used at test time for computing

optimization objectives and adapts the network to the test sample. Entropy minimization is another common technique for developing self-supervised objectives [46, 47]. TENT [48] performs test-time optimization by minimizing the entropy of the batch-wise prediction probability distributions, but it needs more than one test sample to get a non-trivial solution. Zhang et al. [49] propose marginal entropy minimization that works on a single test sample with data augmentations. Another major challenge is to choose the right parameter group for optimization. Batch normalization (BN) layers have been shown to capture the domain discrepancies in image data [50, 51]. It is a straightforward way to directly adapt the BN statistics at test time to enhance model robustness against distribution shifts [52]. However, adapting BN layers puts restrictions on model architectures. Another choice is to update the feature extractor while freezing the prediction module [44, 53]. Zhang et al. [49] show that optimizing the entire model at test time can work as well. Our method addresses both of the challenges above. For the choice of parameter group, we optimize the text prompt while keeping the model intact. Our motivation is to avoid distorting the pre-trained features and to preserve the zero-shot generalization ability of pre-trained models. In section 5, we empirically show that the prompt works as the most effective parameter group for CLIP. Different from the previous single-point method [49], we refine the entropy minimization by proposing *confidence selection*, which helps filter out noisy augmentations that may lead to misleading predictions.

## 3 TPT: Test-Time Prompt Tuning

In this section, we first discuss how to apply CLIP to downstream tasks in a zero-shot manner with a hand-crafted prompt. Next, we briefly review recent progress in prompt tuning approaches for CLIP using downstream training data. Finally, we give detailed introductions of how to apply our method to the image classification task and context-dependent visual reasoning, respectively.

### 3.1 Background

**Contrastive Language-Image Pre-training (CLIP).** CLIP consists of two parallel encoders, one that maps the text input into a feature vector, and the other does the same for the image input. The model is trained with a contrastive loss that promotes similarity between the two vectors so that the text and image align in the feature space. We denote a CLIP model as $\mathcal{F} = \{\mathbf{E}_{\texttt{visual}}, \mathbf{E}_{\texttt{text}}\}$, with $\mathbf{E}_{\texttt{visual}}$ and $\mathbf{E}_{\texttt{text}}$ being the image and text encoders.

We first review how to apply CLIP to downstream tasks in a zero-shot manner with a hand-crafted prompt. We take image classification as an example. Consider a single test image $X_{test}$ of class $y$, where $X \in \mathbb{R}^{C \times H \times W}$ and $y \in \mathbb{R}^K$ for a $K$-class classification problem. In the baseline zero-shot setting, we prepend a hand-crafted prompt prefix, such as $\boldsymbol{p}$ ="a photo of a", to every $y_i$ in $\mathcal{Y} = \{y_1, y_2, \ldots, y_K\}$ to form the category-specific text inputs $\{\boldsymbol{p}; y_i\}$. We then feed these class descriptions to the text encoder to get the text features $\{\boldsymbol{t}_1, \boldsymbol{t}_2, \ldots, \boldsymbol{t}_K\}$, where $\boldsymbol{t}_i = \mathbf{E}_{\texttt{text}}(\{\boldsymbol{p}; y_i\})$. Each text feature $\boldsymbol{t}_i$ is paired with the image feature $\boldsymbol{v} = \mathbf{E}_{\texttt{visual}}(X)$ to compute a similarity score $\boldsymbol{s}_i = \texttt{sim}(\boldsymbol{t}_i \cdot \boldsymbol{v})$, where $\texttt{sim}(,)$ denotes the cosine similarity. The prediction probability on $X$ can be denoted by $p(y_i|X) = \frac{\exp(\texttt{sim}(\boldsymbol{t}_i \cdot \boldsymbol{v})\tau)}{\sum_{i=1}^{K} \exp(\texttt{sim}(\boldsymbol{t}_i \cdot \boldsymbol{v})\tau)}$, where $\tau$ is the temperature of the softmax function.

**Prompt tuning using downstream training data.** Instead of using a hand-crafted prompt, prompt tuning methods train a prompt to maximize performance on a downstream task for which labeled data is available. Prompt tuning optimizes the prompt $\boldsymbol{p} \in \mathbb{R}^{L \times D}$ in the text embedding space, with the number of tokens $L$ and embedding size $D$, using training data with annotations $\mathcal{D}_{\text{train}} = \{(X_i, y_i)\}$ from the downstream task. The goal is to obtain text inputs $\{\boldsymbol{p}; \mathcal{Y}\} = \{\{\boldsymbol{p}; y_i\}$ for $y_i \in \mathcal{Y}\}$ that can provide the model with the most helpful context information about the task. For image classification with cross-entropy loss $\mathcal{L}$, the problem can be formulated as:

$$\boldsymbol{p}^* = \arg\min_{\boldsymbol{p}} \mathbb{E}_{(X,y) \sim \mathcal{D}_{\text{train}}} \mathcal{L}(\mathcal{F}_{\boldsymbol{p}}(X), y), \tag{1}$$

$$\text{where } \mathcal{F}_{\boldsymbol{p}}(X) = \texttt{sim}(\mathbf{E}_{\texttt{text}}(\{\boldsymbol{p}; \mathcal{Y}\}), \mathbf{E}_{\texttt{visual}}(X)). \tag{2}$$

**Context-dependent visual reasoning.** For the task of context-dependent visual reasoning, such as Bongard-HOI [16], a test sample contains two sets of support images and a query image for evaluation. The two sets of support images exemplify the presence and the absence of a human-object interaction (HOI) concept (*e.g.,* "ride bike"). The model is then asked to infer whether the query image contains

the underlying concept. Specifically, each concept in this task is a visual relationship $c = \langle s, a, o \rangle$, with $s$ being the subject ($s =$"human" for HOI tasks), $a$ denoting the action and $o$ for the object. Each test sample $X_{\text{test}}$ captures a concept by presenting $c = \langle s, a, o \rangle$ in one set of support images (positive examples), while having the other set (negative examples) to demonstrate $c' = \langle s, a', o \rangle$, where $a' \neq a$. Note that neither $o$ nor $a$ is given explicitly, and it relies on the model's reasoning ability to predict whether the query image contains the featured concept $c$ of the test sample.

Existing methods [54, 55] approach the Bongard-HOI problem by training the model on a collection of similar tasks (using the Bongard-HOI training split) so that it can make similar inferences on test samples at test time. When applying CLIP to this task, we do not use additional training data because CLIP has learned abundant visual concepts and thus is a natural fit for such visual reasoning tasks.

## 3.2 TPT: Test-Time Prompt Tuning

**Why optimize prompts?** CLIP contains rich knowledge obtained from pre-training on a massive and diverse dataset. However, how to *more effectively* extract such knowledge remains an open question. A simple strategy is to directly fine-tune the model, either end-to-end or for a subset of layers, on a category of inputs. However, previous work has shown that such fine-tuning strategies result in domain-specific behaviors that lose the out-of-distribution generalization and robustness of foundation models [13, 56]. Prompts, on the other hand, work outside the pre-trained model by modifying the context of the model input, thus do not distort pre-trained features.

In this work, our goal is to leverage the existing knowledge of CLIP to boost its generalization in a zero-shot manner. Therefore, prompt tuning serves as an ideal handle to approach the goal. Furthermore, we regard test-time prompt tuning as a way to provide the model with the context tailored to the single test sample, which helps precisely retrieve the knowledge of CLIP.

At the inference stage, the only information available is the single test sample $X_{\text{test}}$ without label information. TPT, therefore, manages to optimize the prompt $\boldsymbol{p}$ at test time based on the single test sample. In general, our objective can be formulated in the form of

$$\boldsymbol{p}^* = \arg\min_{\boldsymbol{p}} \mathcal{L}(\mathcal{F}, \boldsymbol{p}, X_{\text{test}}) \tag{3}$$

for some carefully constructed loss. Note that, unlike equation (1), our method does not require any labels or any data beyond the zero-shot test sample.

**TPT for image classification.** Because labels are not available for test time tuning, we must select an unsupervised loss for prompt tuning. We design our TPT objective to promote the consistency of the model's predictions across different augmented views of a given test image. Specifically, we generate $N$ randomly augmented views of the test image using a family of random augmentations $\mathcal{A}$, and minimize the entropy of the averaged prediction probability distribution:

$$\boldsymbol{p}^* = \arg\min_{\boldsymbol{p}} - \sum_{i=1}^{K} \tilde{p}_{\boldsymbol{p}}(y_i|X_{\text{test}}) \log \tilde{p}_{\boldsymbol{p}}(y_i|X_{\text{test}}), \tag{4}$$

$$\text{where } \tilde{p}_{\boldsymbol{p}}(y_i|X_{\text{test}}) = \frac{1}{N} \sum_{i=1}^{N} p_{\boldsymbol{p}}(y_i|\mathcal{A}_i(X_{\text{test}})). \tag{5}$$

Here, $p_{\boldsymbol{p}}(y|\mathcal{A}_i(X_{\text{test}}))$ is the vector of class probabilities produced by the model when provided with prompt $\boldsymbol{p}$ and the $i$-th augmented view of the test image.

In addition, to reduce the noise from random augmentations, we propose *confidence selection* to filter out views that generate high-entropy (*i.e.*, low-confidence) predictions. Such views of an image may lack important information needed to classify it correctly, *e.g.*, a random crop may have removed important image content. We select confident samples with a prediction entropy below a threshold $\tau$. We adapt $\tau$ for each test sample, by taking the entropy value at the $\rho$-percentile among the self-entropy of $N$ augmented views ranked from low to high (*i.e.*, confidence from high to low). With $\tau$, the confidence selection can be written as a mask over the augmented samples $\mathbb{1}[\mathbf{H}(p_i) \leq \tau]$, with $\mathbf{H}$ measuring the self-entropy of the prediction on an augmented view. Using confidence selection with

a cutoff percentile $\rho$ on $N$ augmented views, the averaged probability in Eq. (4) now becomes:

$$\tilde{p}_{\boldsymbol{p}(y|X_{\text{test}})} = \frac{1}{\rho N} \sum_{i=1}^{N} \mathbb{1}[\mathbf{H}(p_i) \leq \tau] p_{\boldsymbol{p}}(y|\mathcal{A}_i(X_{\text{test}})), \tag{6}$$

**TPT for context-dependent visual reasoning.** Different from image classification, where every image has one and only ground-truth label, the correctness of the prediction in Bongard-HOI depends on the context (*i.e.*, example images), which is binary (containing the concept $c$ or not). In the case of binary labels, a straightforward prompting strategy is to hand-craft "labels" for positive and negative examples, such as "True/False" or "Yes/No". With TPT, on the other hand, we can directly learn an optimal label token $\boldsymbol{cls}$ on the example images in the test sample. More importantly, for visual reasoning, TPT can explicitly learn the context (*i.e.*, visual concept) in the form of text prompts, and assists visual reasoning of vision-language models with language context. Formally, given $M$ support images in each test sample, the TPT objective for context-dependent reasoning can be written as:

$$\boldsymbol{p}^* = \arg\min_{\boldsymbol{p}, \boldsymbol{cls}} \frac{1}{M} \sum_{X \in \{X_{\mathcal{P}}, X_{\mathcal{N}}\}} \mathcal{L}(\mathcal{F}_{\boldsymbol{c}, \boldsymbol{cls}}(X), y), \tag{7}$$

where we assign $y \in \{0, 1\}$ to negative and positive example images respectively for computing the cross-entropy loss $\mathcal{L}$. Unlike in image classification, we tune the binary label tokens $\boldsymbol{cls} = \{\boldsymbol{cls}^1, \boldsymbol{cls}^2\}, \boldsymbol{cls}^i \in \mathbb{R}^{1,D}$ and prompt $\boldsymbol{p} \in \mathbb{R}^{L,D}$ simultaneously. For each image, we assemble the text input to CLIP as $T = \{T_1, T_2 \,|\, T_i = \{\boldsymbol{p}, \boldsymbol{cls}^i\}\}$.

Note that the support set is an essential part of a Bongard-HOI sample, which provides the context for this context-dependent task. Therefore, our approach is still considered to work purely at test time, without training data or annotations (*i.e.*, TPT has not been trained on a collection of similar tasks from the Bongard-HOI training split).

## 4 Experiments

In this section, we describe the tasks and benchmarks used for evaluating our method, along with the implementation details. Our main results cover three aspects of the model's generalization: robustness to natural distribution shifts, cross-dataset generalization, and context-dependent visual reasoning. We also provide ablation experiments in section 5, analyzing different network components for test-time tuning, and other design choices of our method.

### 4.1 Robustness to Natural Distribution Shifts

**Datasets.** CLIP has been shown to be robust to distribution shifts that can occur naturally in real-world scenarios. We follow the setting in Radford et al. [1] and evaluate the model's robustness to natural distribution shifts on 4 ImageNet Variants as follows, which have been considered as out-of-distribution (OOD) data for ImageNet [57] in previous work.

- **ImageNet-V2** [58] is a independent test set containing natural images, collected from different source, including 10,000 images of 1,000 ImageNet categories.
- **ImageNet-A** [59] is a challenging test set of "natural adversarial examples" misclassified by a standard ResNet-50 [60], consisting of 7,500 images of 200 of ImageNet categories.
- **ImageNet-R** [14] collects images of ImageNet categories but with artistic renditions. There are 30,000 images in total, including 200 ImageNet categories.
- **ImageNet-Sketch** [61] is a dataset of black and white sketches, collected independently from the original ImageNet validation set. The dataset includes 50,000 images in total, covering 1,000 ImageNet categories.

**Baselines.** We compare TPT with existing few-shot prompt tuning methods that are designed for CLIP. CoOp [28] is a few-shot prompt tuning baseline that tunes a fixed dataset-specific prompt on each downstream dataset. CoCoOp [15] is the state-of-the-art prompt tuning method for CLIP. It produces input-dependent prompts with a network module, of which the input is the image feature. The network module of CoCoOp is also trained on downstream data in a dataset-specific way.

Following their original configuration, we train both methods on ImageNet using 16-shot training data per category with 4 learnable prompt tokens and directly test the tuned prompt on OOD benchmarks. We also include two versions of the baseline zero-shot performance of CLIP, using a default prompt "a photo of a", and the ensemble of 80 hand-crafted prompts from Radford et al. [1].

**Implementation details.** For TPT, we initialize the prompt as the default hand-crafted one "a photo of a", and optimize the corresponding 4 tokens in the text input embedding space based on a single test image. We augment a single test image 63 times using random resized crops and construct a batch of 64 images, including the original one. Among the 64 predictions, we select the top 10% ($\rho$=0.1) confident samples (lowest 10% in self-entropy) and compute the entropy of the averaged probability of the selected predictions (i.e., marginal entropy). We optimize the prompt to minimize the marginal entropy for 1 step, using the AdamW optimizer with a learning rate of 0.005.

**Results.** In Table 1, the standalone TPT achieves higher accuracy than both prompt ensemble and existing few-shot prompt tuning methods, including CoCoOp. Furthermore, since TPT works at test time, it is complementary to existing baseline methods. We show that by applying TPT to prompts learned by CoOp or CoCoOp, we can further improve the accuracy of their in-domain ImageNet data, as well as generalization ability to OOD data. We also compare TPT to the ensembles of baseline models in Appendix A.3, where we find that applying TPT to baseline methods can bring more substantial improvement than model ensembles. In addition, among the five datasets, few-shot prompt tuning methods bring the most accuracy gain on the ImageNet validation set and ImageNet-V2. However, on datasets with more significant distribution shifts, few-shot prompt tuning methods trained on ImageNet show no advantage over the ensemble of hand-crafted prompts.

Table 1: **Robustness to Natural Distribution Shifts**. CoOp and CoCoOp are tuned on ImageNet using 16-shot training data per category. Baseline CLIP, prompt ensemble, and TPT do not require training data.

| Method | ImageNet Top1 acc. ↑ | ImageNet-A Top1 acc. ↑ | ImageNet-V2. Top1 acc. ↑ | ImageNet-R. Top1 acc. ↑ | ImageNet-Sketch Top1 acc. ↑ | Average | OOD Average |
|---|---|---|---|---|---|---|---|
| CLIP-RN50 | 58.16 | 21.83 | 51.41 | 56.15 | 33.37 | 44.18 | 40.69 |
| Ensemble | 59.81 | 23.24 | 52.91 | **60.72** | 35.48 | 46.43 | 43.09 |
| CoOp | 63.33 | 23.06 | 55.40 | 56.60 | 34.67 | 46.61 | 42.43 |
| CoCoOp | 62.81 | 23.32 | 55.72 | 57.74 | 34.48 | 46.81 | 42.82 |
| TPT | 60.74 | 26.67 | 54.70 | 59.11 | 35.09 | 47.26 | 43.89 |
| TPT + CoOp | **64.73** | **30.32** | **57.83** | 58.99 | **35.86** | **49.55** | **45.75** |
| TPT + CoCoOp | 62.93 | 27.40 | 56.60 | 59.88 | 35.43 | 48.45 | 44.83 |
| CLIP-ViT-B/16 | 66.73 | 47.87 | 60.86 | 73.98 | 46.09 | 59.11 | 57.20 |
| Ensemble | 68.34 | 49.89 | 61.88 | **77.65** | 48.24 | 61.20 | 59.42 |
| CoOp | 71.51 | 49.71 | 64.20 | 75.21 | 47.99 | 61.72 | 59.28 |
| CoCoOp | 71.02 | 50.63 | 64.07 | 76.18 | 48.75 | 62.13 | 59.91 |
| TPT | 68.98 | 54.77 | 63.45 | 77.06 | 47.94 | 62.44 | 60.81 |
| TPT + CoOp | **73.61** | **57.95** | **66.83** | 77.27 | **49.29** | **64.99** | **62.83** |
| TPT + CoCoOp | 71.07 | 58.47 | 64.85 | 78.65 | 48.47 | 64.30 | 62.61 |

## 4.2 Cross-Datasets Generalization

Pre-trained vision-language models like CLIP are ideal for "open-world" problems. For example, we can apply CLIP to classify arbitrary categories in a zero-shot manner in image classification.. However, a prompt tuned on a specific downstream dataset can be less generalizable to categories outside its training set. In this section, we evaluate the cross-dataset generalization of existing few-shot prompt tuning methods (same as in section 4.1), and compare them with TPT, which is not dataset-specific.

**Setup.** We conduct a cross-dataset evaluation on the task of image classification. We consider 10 datasets, covering fine-grained classifications including species of plants or animals (Flower102 [62], OxfordPets [63]), scenes (SUN397 [64]), textures (DTD [65]), food (Food101 [66]), transportation (StanfordCars [67], Aircraft [68]), human actions (UCF101 [69]), satellite images (EuroSAT [70]), and general objects (Caltech101 [71]). We consider two different settings of cross-dataset generalization. In the first setting, we consider ImageNet with 1000 categories as a comprehensive source dataset, and use other fine-grained datasets as target datasets for evaluation. We implement CoOp

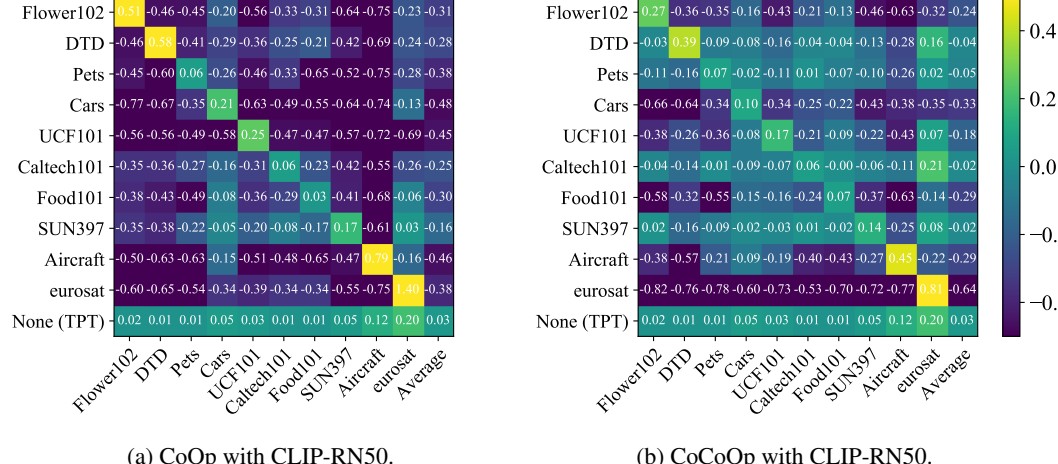

(a) CoOp with CLIP-RN50.  (b) CoCoOp with CLIP-RN50.

Figure 2: **Cross-dataset improvement normalized by the zero-shot baseline performance.** In each matrix $A$, $A_{i,j}$ is the *normalized relative improvement* on the $j_{th}$ dataset of using the prompt tuned on the $i$-th dataset. The value $A_{i,j}$ stands for how well a method trained on a source dataset $i$ performs on a target dataset $j$, in comparison with a zero-shot CLIP baseline (using a hand-crafted prompt). Thus, the higher, the better. The last row is the performance of TPT, which is not tuned on any source dataset. The last column summarizes the average improvement over 10 datasets, measuring the overall generalization ability across the 10 datasets.

and CoCoOp using the same setting as in section 4.1, and evaluate their generalization performance to the 10 datasets. In the second setting, we consider a more challenging scenario, where the source data for few-shot prompt tuning also comes from the specialized fine-grained datasets, and there is no overlapping in categories between a source-target pair.

**Implementation details.** We implement CoOp and CoCoOp on each source dataset following their original configurations. For TPT, we initialize the prompt as "a photo of a" for every datasets. We adopt the same hyper-parameter setting as in section 4.1. We use AugMix [72] as a stronger data augmentation for this task.

Table 2: **Cross-dataset generalization from ImageNet to fine-grained classification datasets.** CoOp and CoCoOp are tuned on ImageNet using 16-shot training data per category. Baseline CLIP, prompt ensemble, and TPT do not require training data or annotations. We report the top-1 classification accuracy on each dataset.

| Method | Flower102 | DTD | Pets | Cars | UCF101 | Caltech101 | Food101 | SUN397 | Aircraft | EuroSAT | Average |
|---|---|---|---|---|---|---|---|---|---|---|---|
| CLIP-RN50 | 61.75 | 40.37 | 83.57 | 55.70 | 58.84 | 85.88 | 73.97 | 58.80 | 15.66 | 23.69 | 55.82 |
| Ensemble | 62.77 | 40.37 | 82.97 | 55.89 | 59.48 | **87.26** | 74.82 | 60.85 | 16.11 | 25.79 | 56.63 |
| CoOp | 61.55 | 37.29 | 87.00 | 55.32 | 59.05 | 86.53 | 75.59 | 58.15 | 15.12 | 26.20 | 56.18 |
| CoCoOp | **65.57** | 38.53 | **88.39** | 56.22 | 57.10 | 87.38 | **76.20** | 59.61 | 14.61 | **28.73** | 57.23 |
| TPT | 62.69 | **40.84** | 84.49 | **58.46** | **60.82** | 87.02 | 74.88 | **61.46** | **17.58** | 28.33 | **57.66** |
| CLIP-ViT-B/16 | 67.44 | 44.27 | 88.25 | 65.48 | 65.13 | 93.35 | 83.65 | 62.59 | 23.67 | 42.01 | 63.58 |
| Ensemble | 66.99 | 45.04 | 86.92 | 66.11 | 65.16 | 93.55 | 82.86 | 65.63 | 23.22 | **50.42** | 64.59 |
| CoOp | 68.71 | 41.92 | 89.14 | 64.51 | 66.55 | 93.70 | **85.30** | 64.15 | 18.47 | **46.39** | 63.88 |
| CoCoOp | **70.85** | 45.45 | **90.46** | 64.90 | **68.44** | **93.79** | 83.97 | **66.89** | 22.29 | 39.23 | 64.63 |
| TPT | 68.98 | **47.75** | 87.79 | **66.87** | 68.04 | 94.16 | 84.67 | 65.50 | **24.78** | 42.44 | **65.10** |

**Results.** In Table 2, we compare TPT with few-shot prompt tuning methods on generalization from ImageNet to fine-grained datasets. Note that TPT works in a zero-shot manner; thus it is not trained on ImageNet. Nonetheless, we find TPT to achieve on-par generalization as ImageNet trained CoCoOp. In Figure 2, we present the results of the more challenging setting of cross-dataset generalization, where there is no overlap between the source and target dataset. For better visualization, we plot the relative accuracy improvement $acc' = (acc - acc_{base})/acc_{base}$, normalized by the zero-shot baseline accuracy $acc_{base}$ of a CLIP-RN50. For example, baseline CLIP with a hand-crafted prompt achieves 61.75% accuracy on Flower102, while CoOp trained on DTD only has 33.41% on Flower102. In

this case, we calculate $acc'$ as $(33.41 - 61.75)/61.75 = -0.46$. From Figure 2, we can see that the averaged accuracy improvement (in the last column of each matrix) of few-shot prompt tuning methods is always negative, meaning that they do worse than the zero-shot baseline. TPT, on the other hand, shows consistent improvement in each of the 10 datasets.

## 4.3 Context-dependent Visual Reasoning on Bongard-HOI

**Baselines.** We include three previous methods for comparison: (1) The CNN-baseline [54] is a simple classifier trained on Bongard-HOI training data, for which the model is trained to map a training sample as a whole (including the support and query images) to a binary output, indicating whether the query image contains the corresponding concept; (2) The Meta-baseline [55] regards each sample in Bongard-HOI as a few-shot task and the model is trained on the Bongard-HOI training data with a meta-objective that aims to quickly adapt the model to new tasks; (3) HOITrans [17] is the previous best method on Bongard-HOI. It is a transformer-based HOI detection model that achieves state-of-the-art accuracy on various HOI detection benchmarks. It solves Bongard-HOI by comparing the detected HOIs of the query images to those of the support images.

Table 3: **Evaluation on the Bongard-HOI benchmark**. CNN and Meta baselines are implemented based on a ResNet-50 (RN50). ($*$ denotes that the method uses ground truth bounding boxes to assist the inference.)

| Method | Test Splits | | | | Average |
| --- | --- | --- | --- | --- | --- |
| | seen act., seen obj., | unseen act., seen obj., | seen act., unseen obj., | unseen act., unseen obj., | |
| CNN-baseline | 50.03 | 49.89 | 49.77 | 50.01 | 49.92 |
| Meta-baseline* | 58.82 | 58.75 | 58.56 | 57.04 | 58.30 |
| HOITrans | 59.50 | 64.38 | 63.10 | 62.87 | 62.46 |
| TPT (w/ CLIP-RN50) | **66.39** | **68.50** | **65.98** | **65.48** | **66.59** |

**Implementation details.** For each Bongard-HOI test sample, TPT tunes the prompt prefix and class tokens simultaneously from scratch. All learnable tokens are initialized in the text embedding space from a Gaussian distribution with $\sigma = 0.02$. We optimize the prompt on all support images of a test sample for 64 steps, using the AdamW optimizer with a learning rate of 0.005, and then infer the query image with the updated prompt. For other baselines, we directly report the results from Jiang et al. [16], and we refer interested readers to this paper for more details. Note that the HOITrans model is trained on all possible HOI concepts, including the ones in the testing splits.

**Results.** In Table 3, we follow the setup in Jiang et al. [16], and compare TPT with previous methods on 4 test splits of Bongard-HOI respectively. In Bongard-HOI, test images are split into four sets by their overlap in the HOI concept with the training data: whether the action $a$ or the object $o$ has appeared in the training data. Note that our CLIP-based TPT is not trained on the training split of Bongard-HOI, and thus the definition of the four splits is not strictly applicable to TPT.

## 5 Ablation Study

In this section, we analyze our design choices and provide ablation study on the effects of different components of TPT. For simplicity, if not otherwise specified, analyses in this section are conducted on the natural distribution shifts benchmarks. We first compare test-time optimization on different parameter groups of CLIP, showing that prompt tuning achieves the most accuracy gain. Next, we show the improvement brought by confidence selection and analyze how the confidence threshold affects performance. Additional ablation studies can be found in Appendix A.3. We also provide a qualitative study on the effect of TPT on model prediction probability distributions in Appendix A.8.

**Test-time optimization on different parameter groups of CLIP.** Existing test-time optimization methods have worked on different parameter groups of a model. Although there is a strong intuition for tuning prompts on CLIP, it is unclear whether it is the most effective choice. In Figure 3 (a), we evaluate the performance of test-time optimization on different parameter groups of CLIP. We consider four different parameter groups: 1) the entire model, 2) the text encoder, 3) the visual encoder, and 4) the text prompt. For a fair comparison, we adopt the same setup as MEMO [49] by

using AugMix [72] as the data augmentation. Confidence selection is not used in this ablation study. For each design choice, we run a grid search for hyper-parameter tuning (on the learning rate and the number of optimization steps) and report the best result.

The result suggests that text prompt is the most effective parameter group. In addition, we find optimizing the visual encoder to have the worst result. This observation is in alignment with previous work that suggests fine-tuning the image encoder can distort pre-trained features [73, 56].

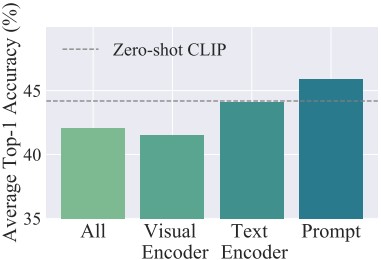

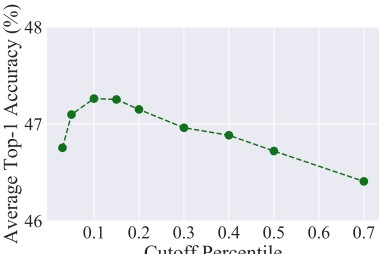

(a) Test-time optimization on different modules.   (b) Different cutoff percentile in confidence selection.

Figure 3: **Ablating the effects of different components of TPT.** We evaluate the top-1 accuracy on the distribution shifts benchmarks in section 4.1. Methods are implemented based on a CLIP-RN50.

**The effect of confidence selection.**    We present confidence selection as a major component of our method, which filters out "noisy" augmented views that provide little information. In Table 4, we provide the performance of TPT without confidence selection, in comparison with the full method. Confidence selection brings non-trivial performance improvement to our baseline TPT. We further show the effect of confidence threshold $\rho$ in Figure 3 (b). The result suggests that using the top-10% confident sample leads to the highest average accuracy. In addition, we find that the effect of confidence selection is generalizable to other entropy-based test-time optimization methods. More details about this analysis are included in appendix A.4

Table 4: **The effect of confidence selection**. The last row is the performance of our full method.

| Method | ImageNet Top1 acc. ↑ | ImageNet-A Top1 acc. ↑ | ImageNet-V2. Top1 acc. ↑ | ImageNet-R. Top1 acc. ↑ | ImageNet-Sketch Top1 acc. ↑ | Average | OOD Average |
|---|---|---|---|---|---|---|---|
| CLIP-RN50 | 58.16 | 21.83 | 51.41 | 56.15 | 33.37 | 44.18 | 40.69 |
| baseline TPT | 60.31 | 23.65 | 53.66 | 57.48 | 34.31 | 45.88 | 42.28 |
| + confidence selection | 60.74 (+0.43) | 26.67 (+3.02) | 54.70 (+1.04) | 59.11 (+1.63) | 35.09 (+0.78) | 47.26 (+1.38) | 43.89 (+1.61) |

# 6   Conclusion

In this work, we investigated how to fully exploit the potential of pre-trained vision-language foundation models as better zero-shot learners. We developed Test-time Prompt Tuning (TPT), a new prompt tuning method that can learn adaptive prompts on the fly with a single test sample. We demonstrated the effectiveness of our method on the robustness to natural distribution shifts and cross-dataset generalization, by using CLIP as the foundation model. Without the need for any training data or annotations, TPT improves the zero-shot generalization ability of CLIP.

**Limitations.** While TPT does not require training data or annotations, our method requires a one-step backpropagation when optimizing the prompt at test time. Since TPT generates multiple augmented views of a single test sample, it increases the memory cost during inference.

**Future directions.** The idea of TPT can be applied to other foundation models for various downstream tasks, including other vision-language models [6, 74] and foundation models of other modalities (*e.g.*, pre-trained large-scale language models [19, 18]) to further boost their zero-shot generalization. The most interesting part of this direction is to design a test-time objective that fits the nature of the model and the downstream task. Besides, it is also interesting to explore how to reduce the memory cost of TPT and make it more computationally efficient.

# 7   Acknowledgements

Shu and Goldstein were supported by the ONR MURI program and DARPA GARD.

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
