# A Appendix

## A.1 Broader Impact

We introduce TPT to promote the generalization of vision-language foundation models. We believe it is worth exploring the zero-shot robustness and generalization of large-scale foundation models as they present a promising direction toward more reliable machine learning systems. In this work, to better leverage the knowledge of pre-trained foundation models, we explore the idea of prompt tuning because it does not alter the inner representations of pre-trained models. We hope this work inspires future studies in reaching the full potential of foundation models, especially in their robustness and generalization.

## A.2 A diagram of applying TPT to context-dependent visual reasoning (Bongard-HOI)

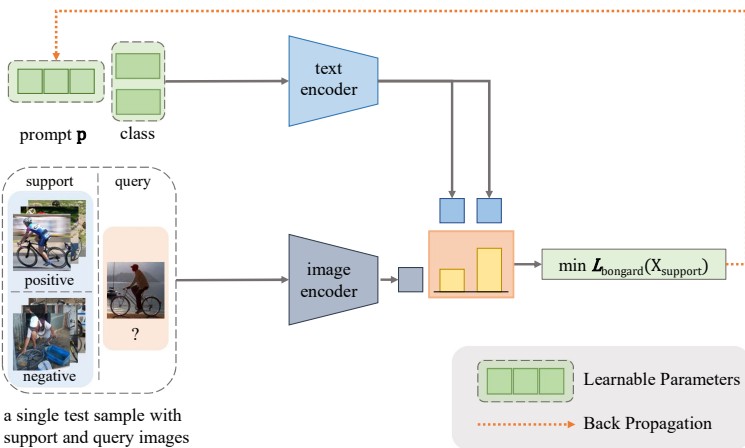

Figure 4: **Test-time Prompt Tuning (TPT) for context-dependent visual reasoning on Bongard-HOI benchmark.** A test sample in Bonagrd-HOI consists of several support images that exemplify a visual concept, and the model needs to predict whether the query image contains the concept. TPT tunes the prompt and class tokens simultaneously on the support images using the cross-entropy loss.

## A.3 More Experiments.

**The trade-off between inference efficiency and accuracy.** We analyze two factors that affect TPT's efficiency: 1) The number of augmented views $N$ that increases the actual number of test samples; 2) The number of optimization steps that increases the runtime and memory usage mainly induced by backpropagation. Figure 5 shows the relationship between the two factors and the average accuracy of TPT on natural distribution shifts.

In Figure 5 (a), the accuracy increases as the number of augmented views grows until reaching a plateau at around $N = 64$. Even when $N = 8$, TPT still brings about over 2% average accuracy gain to the zero-shot CLIP, suggesting that TPT can be adapted for more efficient applications. In Figure 5 (b), we find that increasing the number of optimization steps from 1 to 2 can slightly increase the accuracy (by 0.4%), while there is no significant performance gain from taking more than 2 steps. Considering that the performance gain comes at the expense of linearly increasing the inference time, we use 1-step TPT as our default setting, which is already capable of boosting the average accuracy of zero-shot CLIP by more than 3%.

**Analysis on error bars.** We run TPT multiple times using 3 different random seeds and report the average accuracy with standard deviation. The randomness of TPT mainly comes from random data augmentation and one-step optimization. In addition, we report error bars for few-shot prompt tuning methods (CoOp and CoCoOp). The randomness of these methods mainly comes from the random few-shot sampling of the training data. We provide error bar analysis for the results in Table 1 and Table 2 in the main paper.

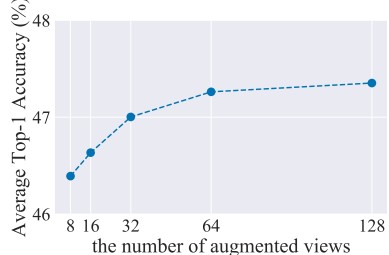

(a) Different number of augmented views.

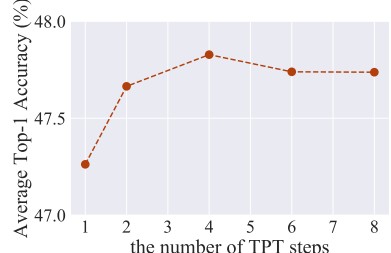

(b) Different number of optimization steps.

Figure 5: **Analysis on the trade-off between efficiency and accuracy.** We evaluate the top-1 accuracy on the distribution shifts benchmarks in section 4.1. Results are based on a CLIP-RN50.

Table 5: **Robustness to natural distribution shifts.** We report the accuracy with an error bar (standard deviation) obtained from three runs with different random seeds.

| Method | ImageNet Top1 acc. ↑ | ImageNet-A Top1 acc. ↑ | ImageNet-V2. Top1 acc. ↑ | ImageNet-R. Top1 acc. ↑ | ImageNet-Sketch Top1 acc. ↑ | Average | OOD Average |
|---|---|---|---|---|---|---|---|
| CLIP-RN50 | 58.16 | 21.83 | 51.41 | 56.15 | 33.37 | 44.18 | 40.69 |
| CoOp | **63.27** (± .07) | 23.23 (± .19) | 55.50 (± .09) | 57.08 (± .42) | 34.68 (± .03) | 46.75(± .12) | 42.62 (± .17) |
| CoCoOp | 62.86 (± .11) | 23.38 (± .50) | **55.59** (± .14) | 57.55 (± .23) | 34.74 (± .29) | 46.82 (± .21) | 42.82 (± .23) |
| TPT | 60.77 (± .03) | **26.60**(± .13) | 54.70 (± .11) | **59.08** (± .03) | **35.17** (± .08) | **47.27** (± .004) | **43.89** (± .003) |
| CLIP-ViT-B/16 | 66.73 | 47.87 | 60.86 | 73.98 | 46.09 | 59.11 | 57.2 |
| CoOp | **71.71** (± .19) | 49.99 (± .29) | **64.49** (± .39) | 75.51 (± .26) | 48.10 (± .14) | 61.96 (± .25) | 59.52 (± .26) |
| CoCoOp | 70.70 (± .32) | 50.76 (± .13) | 63.93 (± .19) | 76.09 (± .29) | **48.60** (± .38) | 62.02 (± .20) | 59.85 (± .19) |
| TPT | 68.96 (± .03) | **54.47** (± .26) | 63.46 (± .07) | **77.10** (± .04) | 47.93 (± .03) | **62.38** (± .05) | **60.74** (± .06) |

From Table 5, our conclusion for remains the same as in Table 1, that TPT achieves higher accuracy than few-shot prompt tuning methods on OOD benchmarks. In addition, we find that on average TPT has a smaller standard deviation than other methods.

In Table 6, we provide the same error bar analysis for the cross-dataset generalization experiment, corresponding to Table 2 in Section 4. Similar to the observation above, our conclusion for cross-dataset evaluation remains the same. On average, we find TPT to have a smaller standard deviation than few-shot prompt tuning methods.

**Comparison with model ensembles.** We include more comparisons with conventional model ensembles of the baselines in Table 1. Note that there exists a fundamental difference between our "ensemble" (e.g., TPT + CoOp) and conventional model ensemble (e.g., naive "hand-crafted" ensemble, CoCoOp with different seeds, etc.). As our method (TPT) works solely at test time and uses a pre-defined prompt (which may come from a hand-crafted prompt, CoOp, or CoCoOp) as the initialization, it is complementary to CoOp, CoCoOp, and hand-crafted prompts.

Specifically, taking "TPT + CoCoOp" as an example, it is done in the following steps: (1). Using the prompt output by CoCoOp as the prompt initialization; (2). Running TPT to tune this prompt at test time to get the final result. However, conventional model ensembles aggregate the predictions obtained from different prompts.

As shown in the table above, ensembles of existing baselines (e.g., CoOp + CoCoOp) fail to bring improvement as substantial as their combination with TPT (e.g. CoOp + TPT or CoCoOp + TPT).

**Other baselines using data augmentation.** To ablate the contribution of data augmentation in our method, we include two additional baselines that are based on data augmentations without any optimization (and without TPT). The *averaged prediction* baseline is to run inference on the augmented images and then average the probability. The *majority vote* baseline runs the majority vote in the predictions of augmented images and takes the results as the final prediction.

We find that both methods fail to achieve comparable improvement as TPT. It suggests that it is non-trivial to design an algorithm for using augmented images.

Table 6: **Cross-dataset generalization from ImageNet to fine-grained classification datasets**. CoOp and CoCoOp are tuned on ImageNet using 16-shot training data per category. Baseline CLIP and TPT do not require training data or annotations. We report the averaged top-1 accuracy on each dataset, with standard deviations obtained from three runs using different random seeds.

| Method | Flower102 | DTD | Pets | Cars | UCF101 | Caltech101 | Food101 | SUN397 | Aircraft | EuroSAT | Average |
|---|---|---|---|---|---|---|---|---|---|---|---|
| CLIP-RN50 | 61.75 | 40.37 | 83.57 | 55.70 | 58.84 | 85.88 | 73.97 | 58.80 | 15.66 | 23.69 | 55.82 |
| CoOp | 61.62 (± .2) | 37.77 (± .9) | 87.24 (± .2) | 55.72 (± .8) | 59.89 (± .8) | 87.23 (± .6) | 75.86 (± .2) | 59.28 (± .9) | 15.20 (± .4) | 25.43 (± 4) | 56.52 (± .7) |
| CoCoOp | **65.11** (± 1) | 39.14 (± .7) | **87.83** (± .6) | 56.40 (± .3) | 58.57 (± 1) | 86.95 (± .5) | **76.18** (± .5) | 60.62 (± .9) | 15.13 (± .5) | **28.79** (± .9) | 57.47 (± .2) |
| TPT | 62.80 (± .3) | **41.43** (± .5) | 84.42 (± .1) | **58.53** (± .1) | **60.64** (± .3) | 87.23 (± .2) | 75.02 (± .1) | **61.46** (± .0) | **17.60** (± .4) | 28.46 (± .1) | **57.76** (± .1) |
| CLIP-ViT-B/16 | 67.44 | 44.27 | 88.25 | 65.48 | 65.13 | 93.35 | 83.65 | 62.59 | 23.67 | 42.01 | 63.58 |
| CoOp | 68.25 (± 0.5) | 42.34 (± 2) | 89.38 (± .2) | 63.35 (± 1) | 67.17 (± 1) | 92.82 (± .5) | 83.74 (± .4) | 64.51 (± .6) | 19.99 (± 2) | 40.22 (± 4) | 63.18 (± .7) |
| CoCoOp | **71.59** (± .6) | 45.48 (± .2) | **90.20** (± .2) | 65.17 (± .2) | **68.77** (± .8) | **94.15** (± .3) | **84.83** (± 1) | **67.07** (± .3) | 22.95 (± .7) | 42.13 (± 3) | **65.23** (± .6) |
| TPT | 68.79 (± .1) | **46.79** (± .1) | 87.09 (± .1) | **66.38** (± .2) | 67.86 (± .1) | 94.13 (± .1) | 84.67 (± .1) | 65.41 (± .1) | **23.44** (± .3) | **42.78** (± .3) | 64.73 (± .1) |

Table 7: **Robustness to Natural Distribution Shifts.**. CoOp and CoCoOp are tuned on ImageNet using 16-shot training data per category. Baseline CLIP, prompt ensemble, and TPT do not require training data.

| Method | ImageNet Top1 acc. ↑ | ImageNet-A Top1 acc. ↑ | ImageNet-V2. Top1 acc. ↑ | ImageNet-R. Top1 acc. ↑ | ImageNet-Sketch Top1 acc. ↑ | Average | OOD Average |
|---|---|---|---|---|---|---|---|
| CLIP-RN50 | 58.16 | 21.83 | 51.41 | 56.15 | 33.37 | 44.18 | 40.69 |
| Hand-crafted ensemble | 59.81 | 23.24 | 52.91 | **60.72** | 35.48 | 46.43 | 43.09 |
| CoOp | 63.33 | 23.06 | 55.40 | 56.60 | 34.67 | 46.61 | 42.43 |
| CoOp (ensemble 3 seeds) | 61.66 | 22.96 | 54.14 | 57.89 | 34.94 | 46.32 | 42.48 |
| CoOp + hand-crafted ensemble | 63.60 | 23.23 | 55.63 | 57.07 | 34.84 | 46.87 | 42.69 |
| CoCoOp | 62.81 | 23.32 | 55.72 | 57.74 | 34.48 | 46.81 | 42.82 |
| CoCoOp (ensemble 3 seeds) | 63.34 | 24.27 | 56.12 | 58.24 | 35.46 | 47.49 | 43.52 |
| CoCoOp + hand-crafted ensemble | 63.03 | 24.16 | 55.73 | 57.88 | 35.22 | 47.20 | 43.25 |
| CoCoOp + CoOp | 63.86 | 23.69 | 56.45 | 57.7 | 35.5 | 47.44 | 43.34 |
| TPT (ours) | 60.74 | 26.67 | 54.7 | 59.11 | 35.09 | 47.26 | 43.89 |
| TPT + CoOp | **64.73** | **30.32** | **57.83** | 58.99 | **35.86** | **49.55** | **45.75** |
| TPT + CoCoOp | 62.93 | 27.4 | 56.6 | 59.88 | 35.43 | 48.45 | 44.83 |

## A.4 Apply Confidence Selection to Other Methods

In section 5, we show that confidence selection is an important part of TPT that improves the baseline entropy minimization. In this section, we further show that confidence selection can also benefit other entropy-based test-time optimization methods, which work on different model architectures and optimize different parameter groups. MEMO [49] minimizes the marginal entropy of the model's predictions across augmented views at test time, by adapting all parameters of a network model.

In Table 9, we implement MEMO based on a standard ResNet-50 from PyTorch, following their original hyper-parameter configurations. To apply confidence selection to a method, we try different cutoff percentile $\rho$ that controls the threshold for confidence selection (smaller $\rho$ means a higher confidence threshold). We find MEMO can also benefit from confidence selection and the improvement increases as the threshold increases (*i.e.* $\rho$ becomes smaller).

## A.5 License information of the assets used in this work.

**Datasets.** Below are the datasets used in this paper that have known license information:
The following datasets used in this paper are under the MIT License: ImageNet-A [59], ImageNetV2 [58], ImageNet-R [14], ImageNet-Sketch [61].
The following datasets used in this paper are under the CC BY-SA 4.0 License: Oxford-IIIT Pets [63].
The following datasets used in this paper are for research purposes only (term of access): ImageNet [57], DTD [65], StanfordCars [67], SUN397 [64], FGVC-Aircraft [68].

**Source code.** We use the implementation of existing baseline methods for reporting their results in this paper. Below are their license information: Source code used in this paper that are under the MIT License: CLIP [1], CoOp [28], CoCoOp [15], TENT [48].

Table 8: **Out-of-distribution evaluation of data augmentation baselines**. We include two additional baselines that are based on data augmentations without any optimization (and without TPT). The limited performance of these baselines suggests that it is non-trivial to design an algorithm for using augmented images.

| Method | ImageNet Top1 acc. ↑ | ImageNet-A Top1 acc. ↑ | ImageNet-V2. Top1 acc. ↑ | ImageNet-R. Top1 acc. ↑ | ImageNet-Sketch Top1 acc. ↑ | Average | OOD Average |
|---|---|---|---|---|---|---|---|
| CLIP-RN50 | 58.16 | 21.83 | 51.41 | 56.15 | 33.37 | 44.18 | 40.69 |
| averaged prediction | 57.94 | 22.71 | 52.02 | 51.25 | 29.51 | 42.69 | 38.87 |
| majority vote | 59.53 | 25.67 | 53.56 | 54.77 | 32.27 | 45.16 | 41.57 |
| TPT (Ours) | **60.74** | **26.67** | **54.70** | **59.11** | **35.09** | **47.26** | **43.89** |

Table 9: **Apply confidence selection to other test-time methods**. All methods are based on a standard ResNet-50. $\rho$ denotes applying confidence selection to a method with a cutoff percentile $\rho$.

| Method | ImageNet Top1 acc. ↑ | ImageNet-A Top1 acc. ↑ | ImageNet-V2. Top1 acc. ↑ | ImageNet-R. Top1 acc. ↑ | ImageNet-Sketch Top1 acc. ↑ | Average | OOD Average |
|---|---|---|---|---|---|---|---|
| ResNet-50 | 76.13 | 0.00 | 63.20 | 36.17 | 24.09 | 39.92 | 30.87 |
| MEMO | 77.23 | 0.75 | 65.03 | 41.34 | 27.72 | 42.41 | 33.71 |
| MEMO ($\rho = 0.7$) | 77.56 | 0.92 | 65.51 | 41.93 | 28.20 | 42.82 | 34.14 |
| MEMO ($\rho = 0.5$) | **77.72** | 1.15 | 65.77 | 42.29 | **28.55** | 43.10 | 34.44 |
| MEMO ($\rho = 0.3$) | 77.57 | 1.43 | **65.85** | 42.64 | 28.33 | 43.16 | 34.56 |
| MEMO ($\rho = 0.1$) | 77.38 | **2.59** | 65.37 | **42.90** | 28.04 | **43.26** | **34.72** |

## A.6  An overview of different prompting strategies for CLIP.

In addition to our discussion on related work in Section 2, we summarize the differences between existing prompting strategies for CLIP in Table 10. We focus on three preferred properties of a prompting strategy and use them to categorize the methods.

Table 10: **An overview of different prompting strategies for CLIP**. "learnable" means the prompt is optimized based on certain objective functions. "no training data" means that no additional data are needed for tuning the prompt. "input-dependent" means the prompt is adaptive to each input instance.

| Prompting Method | learnable | no training data | input-adaptive |
|---|---|---|---|
| Hand-crafted [1] | - | ✓ | - |
| CoOp [28] | ✓ | - | - |
| CoCoOp [15] | ✓ | - | ✓ |
| TPT (ours) | ✓ | ✓ | ✓ |

**Comparison of training and inference budgets.**    We include a comparison of training and inference budgets for prompting strategies of CLIP.

The major computation overhead of TPT comes from the 1-step optimization, which involves back propagations through the text encoder of CLIP. Another overhead comes from the data augmentation, which can be parallelized with little memory increase, as the CLIP's image branch does not require backpropagation. However, note that TPT works solely at test time, and therefore it does not have any training budget. In addition, as shown by our empirical results, prompt tuning without training data can generalize better to unseen distributions.

## A.7  Reproducibility

The code for experiments in this paper can be found here: `https://tinyurl.com/yr3zmhma`. (The code is also included in the supplemental materials.)

For evaluation on natural distribution shifts, we use a single V100 GPU with 32GB memory. For all other experiments, we use a single V100 GPU with 16GB memory.

Table 11: **Comparison of training and testing budgets of prompting strategies for CLIP**.

|  | Hand-crafted [1] | CoOp [28] | CoCoOp [15] | TPT (Ours) |
|---|---|---|---|---|
| Inference speed (seconds/iter) | 0.10 | 0.10 | 0.11 | 0.25 |
| Number of training samples | 0 | 16K | 16K | 0 |
| Number of training iterations | 0 | 12.5K | 800K | 0 |
| Number of additional parameters | 0 | 0 | 65.8K | 0 |

**Details of hyper-parameter tuning.** For the OOD experiments on ImageNet variants, we follow the setting from [15]. We find the optimal set of hyper-parameters on the original ImageNet validation set (not including any of the out-of-distribution data). For the fine-grained image classification, each dataset comes with a validation split and a testing split, and we use the former for hyperparameter tuning. Among the 10 fine-grained datasets, we choose a set of hyper-parameter that achieves the best average validation accuracy.

## A.8 Qualitative Analysis

In this section, we provide a qualitative analysis of the effect of TPT on the probability distribution of the augmented views of a test sample. In each figure, the top left panel shows the test image, and the bar plot on the top right shows CLIP's prediction on the test image before (on the left) and after (on the right) we apply TPT. The bottom two panels show the probability distribution among 200 classes (of ImageNet-R) of 64 augmented views. Each peak of a prediction curve indicates a high probability in the corresponding class. The prediction probability of the original test sample is at the bottom, and we mark the index of the ground-truth class on the x-axis. From the example images, we find that TPT can effectively make the predictions more consistent across different augmented views.

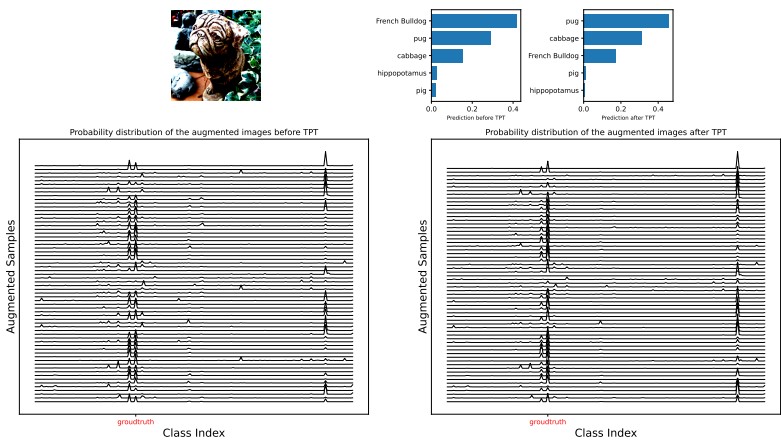

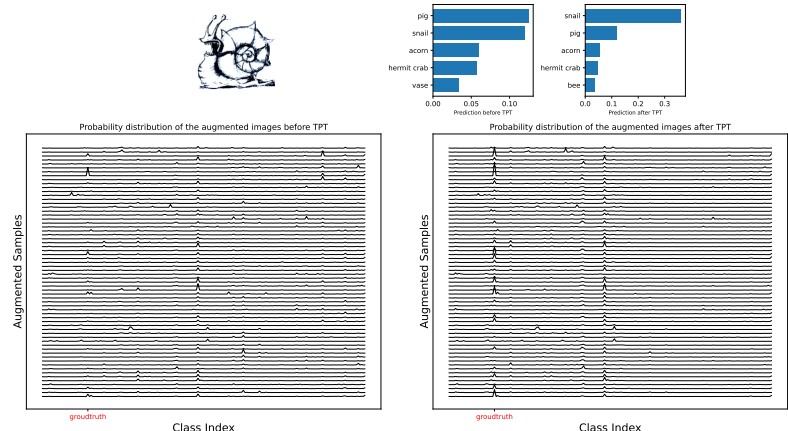

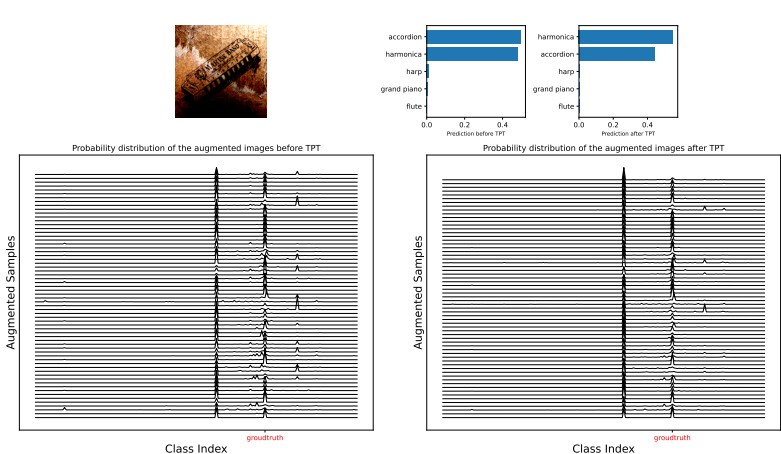

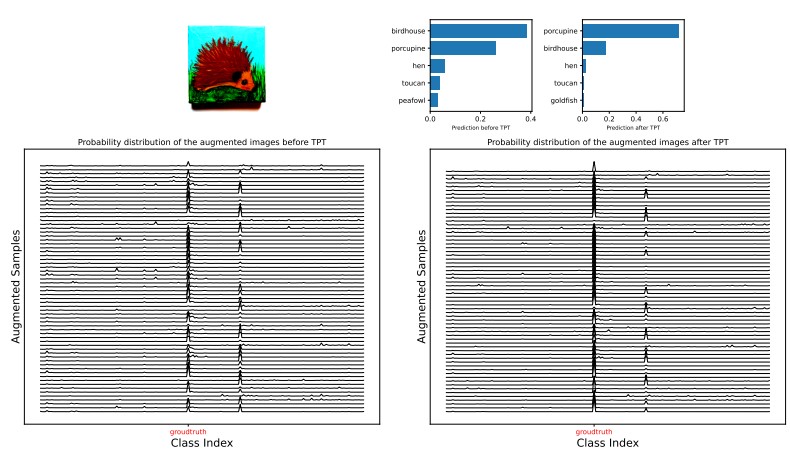