# OpenReview forum: "Test-Time Prompt Tuning for Zero-Shot Generalization in Vision-Language Models"
_NeurIPS.cc/2022/Conference — NeurIPS 2022 Accept_

### Official Review · Reviewer_wmVL · 2022-07-10

**Rating:** 5
**Confidence:** 4
**Soundness:** 3 good
**Presentation:** 4 excellent
**Contribution:** 2 fair

**Summary:**

The authors introduce test-time prompt tuning to improve CLIP's zero-shot classification accuracy by optimizing the consistency of probability distributions across multiple augmented views. To reduce noise samples from random augmentation, the authors proposed confidence selection to remove samples with low confidence. Numerous experiments show the robustness and generalisation of the proposed method.

**Questions:**

1. Compared to the baseline (CLIP, COOP, COCOOP), how much time complexity does the proposed approach increase while improving performance?

2. I think that the optimization objective function (4) proposed by the authors is not reasonable, as I discussed before. Please correct me if I have misunderstood.


**Limitations:**

The authors have discussed the limitations of the proposed approach. I do not see the potential social impact.

**Strengths And Weaknesses:**

Strengths:
1. The proposed method improves the performance of the baselines.
2. This paper is well-organized and easy to follow.

Weaknesses:
1. Although the authors have presented the inference time as a limitation, the paper still lacks a comparison with the time complexity of CLIP, COOP, etc.
2. The authors' motivation is reasonable, but the proposed optimization objective is not. I agree that the predictions should be consistent across perspectives, but it doesn't make sense to keep the predicted class probability vectors consistent. Predictions of non-correct label positions in the probability vector are irrelevant, and it is reasonable to be inconsistent across perspectives.
3. This paper lacks significant contribution and the improvement results from data augmentation.

---

> ### Author Response · Authors · 2022-08-02
> **Reply to Reviewer wmVL**
>
> We thank the reviewer for the feedback and suggestions. We answer your questions below.
>
> > 1. Comparison of the time complexity.
>
> Thanks for the suggestion. We show the comparison below and also include it in Appendix A.6 in our revision. The analysis is conducted on ImageNet-A using a V100 GPU. In addition to the inference speed, we also include the comparison of other budgets.
>
> |                                 | CLIP | CoOp | CoCoOp | TPT |
> |---------------------------------|:----:|:----:|:------:|:---:|
> | inference speed (second/iter)  |  0.10  |  0.10  |  0.11   |  0.25   |
> | number of training iterations       |   0  |  12.5K |   800K   |  0  |
> | number of training samples      |   0  |  16K |  16K   |  0  |
> | number of additional parameters |   0  |   0  |    65.8K    |  0   |
>
> The major computation overhead of TPT comes from the 1-step optimization, which involves backpropagations through the text encoder of CLIP. Another overhead comes from the data augmentation, which can be parallelized with little memory increase, as the CLIP's image branch does not require backpropagation.
>
> However, we would like to emphasize that our method works solely at test time, and therefore it does not have any training budget. In addition, as shown by our empirical results, prompt tuning without training data can generalize better to unseen distributions.
>
> > 2. "Predictions of non-correct label positions in the probability vector are irrelevant, and it is reasonable to be inconsistent across perspectives."
>
> Since TPT is a test-time method, we could not access the label information and thus cannot precisely identify the "non-correct" label positions during optimization. Nonetheless, we agree that should reduce the impact of irrelevant positions, and our optimization objective indeed takes this into account.
>
> Note that TPT minimizes the entropy of the ***averaged*** probability distribution, it allows in-consistency between each individual probability vector. (An example objective of forcing consistency among all positions, on the other hand, would be directly minimizing the cosine distance between the prediction vectors.)
>
> To further explain our statement, we take a toy example for demonstration. For a 3-way classification problem, let $p_a$ = [0.1, 0.3, 0.6] and $p_b$ = [0.3, 0.1, 0.6] be the probability vector of two augmented views of the same image. Their average probability is $\tilde{p}$ = [0.2, 0.2, 0.6], which equals to the averaged probability of an identical pair of probability vectors $p_c$ = [0.2, 0.2, 0.6] and $p_d$ = [0.2, 0.2, 0.6]. It also equals to the average of a more in-consistent pair, such as $p_e$ [0.1, 0.4, 0.5] and $p_f$ = [0.3, 0.0, 0.7], while all of these pairs have the same ***entropy*** of the ***averaged probability***. Therefore, by minimizing the entropy of $\tilde{p}$, we are not forcing each individual probability vector to be equal in every position.
>
> In addition, we also provide a qualitative analysis of the effect of TPT on the probability distribution among augmented images. Take [this image](https://ibb.co/F45190B) as an example, we can see that after TPT tuning, the individual probability distributions remain inconsistent across different augmented images. More examples can be found in Appendix A.8.
>
> > 3. "The improvement results from data augmentation".
>
> As we design our method for the practical test-time setting, we only assume access to a *single test sample* at a time. With the single test sample, we optimize the prompt using a self-supervised objective based on augmentation views of the test sample. However, it is highly non-trivial to design an effective way of using data augmentation in our case, as shown in the table below. We tried two alternative ways of using data augmentations: (1) simply averaging the predictions of all augmented images, and (2) using the majority vote among the augmented images as the final prediction. Both methods fail to achieve comparable improvement as TPT.
>
> |                                        |  ImageNet | ImageNet-A | ImageNet-V2 | ImageNet-R | ImageNet-Sketch |  Average  | OOD Average |
> |----------------------------------------|:---------:|:----------:|:-----------:|:----------:|:---------------:|:---------:|:-----------:|
> | CLIP-RN50                              |   58.16   |    21.83   |    51.41    |    56.15   |      33.37      |   44.18   |    40.69    |
> | average of data augmentations          |   57.94   |    22.71   |    52.02    |    51.25   |      29.51      |   42.69  |   38.87   |
> | majority vote with data augmentations |   59.53   |    25.67   |    53.56    |    54.77   |      32.27      |   45.16   |   41.57   |
> | TPT (Ours)                             | **60.74** |  **26.67** |  **54.70**  |  **59.11** |    **35.09**    | **47.26** |  **43.89**  |

---

> > ### Author Response · Authors · 2022-08-09
> > **Kindly look forward to your feedback**
> >
> > We deeply appreciate your time and constructive feedback. We sincerely hope to see if we have resolved your concerns since the discussion period is ending within 24 hours.
> >
> > In our response above, we include a time complexity analysis as you suggested. We also provide a detailed discussion to answer your question regarding our optimization objective. In addition, we include new results of two data augmentation baselines to show that it is non-trivial to design a method to effectively use the augmented images.
> >
> > We would appreciate the opportunity to provide further information or clarification if needed. We would also appreciate it if you consider raising your rating of this submission.

---

### Official Review · Reviewer_D5Y3 · 2022-07-10

**Rating:** 6
**Confidence:** 4
**Soundness:** 3 good
**Presentation:** 3 good
**Contribution:** 3 good

**Summary:**

This paper proposes a test-time tuning method that adapts the pretrained CLIP for each test example using the example itself.
Such tuning is conducted with the objective of encouraging consistent predictions for different augmented views of the test image, and only soft prompts are updated during this process. The authors demonstrate the effectiveness of this method for image classification task under natural distribution shifts setting and cross-dataset generalization setting, outperforming existing methods that even use additional image classification data for training.



**Questions:**

- The method uses multiple augmented images for one classification, and thus can be regarded as an ensemble method. Did the author compare with a baseline that runs inference on these images and then average the probability? Basically, if we have those augmented images, are there other methods to use them without the optimization step?
- The 1-step optimization at test time works surprisingly well. I am wondering if the author observed any unstableness of this optimization?

**Limitations:**

The author discussed the limitations pretty well.

**Strengths And Weaknesses:**

Strengths:
- The general idea of test-time tuning by encouraging consistency of different augmented views is novel.
- The performance is surprisingly good, outperforming strong baselines in the two generalization settings.
- The analysis on what parameter groups to optimize at test time is inspiring in that it finds prompt tuning is more effective than tuning other set of parameters or the entire model.
- It's also surprising that only one step of optimization can increase the performance stably.

Weaknesses:
- Although the paper demonstrate the empirical improvement of the proposed method, it's not very clear why encouraging the consistency can lead to such much improvement. I would suggest the author discuss more about the underlying assumption.
- Despite that only one step of backpropagation is required for this test-time training, the method still requires computation on multiple augmented images (8-64), which significantly increases the inference computation (roughly 8-64 x 2 slower).
- The proposed test-time tuning involves several hyper-paramters to be tuned, such as the optimization step or the confidence selection interval. However, it's not clear how they are tuned, which raises the concern of using test data for selecting models.

---

> ### Author Response · Authors · 2022-08-02
> **Reply to Reviewer D5Y3**
>
> We thank the reviewer for the feedback and suggestions. We included new experiments and more details to address your concern.
>
> > 1. "It's not very clear why encouraging the consistency can lead to such much improvement. I would suggest the author discuss more about the underlying assumption."
>
> We leverage an assumption that a robust model's decision boundaries should lie in low-density data regions ([Chapelle & Zien, 2005](http://proceedings.mlr.press/r5/chapelle05b/chapelle05b.pdf)). Consistency-regularization-based methods could achieve this goal by making the network outputs invariant to small input perturbations (e.g., augmentations). The idea of encouraging consistency has been explored in many research areas, such as self-supervised learning (SimCLR ([Chen et al, 2020](https://arxiv.org/pdf/2002.05709.pdf)) and BYOL ([Grill et al., 2020](https://arxiv.org/pdf/2006.07733.pdf))).
> In our work, we use consistency regularization as our test-time prompt tuning objective with the confidence selection module. We assume that an optimal prompt should lead to robust feature representation regardless of different data augmentation views. We provide a qualitative study to demonstrate the effect of TPT on the prediction distribution of augmented images. [This example](https://ibb.co/F45190B) shows that TPT can make prediction distributions more consistent. More examples can be found in Appendix A.8.
> Thanks for the suggestions, we have also included this discussion in the Section 2 of our revision.
>
> > 2. How the hyper-parameters are tuned.
>
> For the OOD experiments on ImageNet variants, we follow the setting from CoOp and CoCoOp. We tune the hyperparameters on the original ImageNet validation set (not including any of the out-of-distribution data). For the fine-grained image classification, each dataset comes with a validation split and a testing split, and we use the former for hyperparameter tuning. Among the 10 fine-grained datasets, we choose a set of hyper-parameter that achieves the best average validation accuracy.
>
> > 3. A baseline that runs inference on the augmented images and then average the probability.
>
> Thanks for your suggestion. We conducted a similar analysis earlier for this work. We include the results below, and in Appendix A.3 in our revision. The **average of augmentations** is to run inference on the augmented images and then average the probability. We find that without tuning the text prompt, simply ensembling the augmented images will decrease the accuracy in some domains.
> Additionally, we come up with another baseline to better use the augmented images without optimization: majority vote (named **Majority vote with data augmentation**）. We run a majority vote in the predictions of augmented images, and take the results as the final prediction. This method shows improvement on the original CLIP, but it does not compete well with TPT.
>
> |                                        |  ImageNet | ImageNet-A | ImageNet-V2 | ImageNet-R | ImageNet-Sketch |  Average  | OOD Average |
> |----------------------------------------|:---------:|:----------:|:-----------:|:----------:|:---------------:|:---------:|:-----------:|
> | CLIP-RN50                              |   58.16   |    21.83   |    51.41    |    56.15   |      33.37      |   44.18   |    40.69    |
> | average of data augmentations          |   57.94   |    22.71   |    52.02    |    51.25   |      29.51      |   42.69  |   38.87   |
> | majority vote with data augmentations |   59.53   |    25.67   |    53.56    |    54.77   |      32.27      |   45.16   |   41.57   |
> | TPT (Ours)                             | **60.74** |  **26.67** |  **54.70**  |  **59.11** |    **35.09**    | **47.26** |  **43.89**  |

---

> > ### Author Response · Authors · 2022-08-09
> > **kindly look forward to your feedback**
> >
> > We appreciate your time and effort in providing constructive comments and suggestions.
> > In our response above, we include additional results and discussions to answer your questions.
> >
> > As we are approaching the end of the discussion period, we sincerely hope to get your feedback to make sure that we have resolved your concerns. We are happy to provide further clarifications if needed. Thanks!

---

### Official Review · Reviewer_Tt1G · 2022-07-11

**Rating:** 7
**Confidence:** 3
**Soundness:** 3 good
**Presentation:** 3 good
**Contribution:** 3 good

**Summary:**

This paper proposes test-time prompt tuning (TPT), a method that can learn adaptive prompts on the fly with a single test sample. It optimizes the prompt by minimizing entropy. The authors show TPT can improve 0-shot performance for classification.

**Questions:**

I'm not sure if the four ImageNet variants are really OOD data? What about other tasks?

**Strengths And Weaknesses:**

### Pros
- The method achieves good performance compared to other language-vision prompting methods
- The method shows a new way to exploit a text-image model like CLIP for mono-modal tasks.
- This method doesn't require additional data or supervision.

### Cons
- The 0-shot performance is still behind the fine-tuning by a large margin (although understandable)
- The idea is not completely novel since consistency regularization has been explored before.

---

> ### Author Response · Authors · 2022-08-02
> **Reply to Reviewer Tt1G**
>
> We appreciate the positive feedback from the reviewer. We address the raised concerns and questions below.
>
> > 1. "The idea is not completely novel since consistency regularization has been explored before."
>
> We agree that consistency regularization is a commonly-used technique, for example, in self-supervised learning and the training of deep generative models. However, our novelty mainly lies in the following two aspects: (1) we consider consistency regularization in the context of how to achieve a better generalization of the vision-language foundation model (i.e., CLIP) in a zero-shot manner. To achieve this goal, we are the first to perform prompt tuning on a single image with test-time optimization, by leveraging consistency regularization as an unsupervised signal. (2) Our ***confidence selection*** for test-time optimization is novel. Apart from other test-time optimization methods, we introduce "confidence selection" as a plug-and-play module to boost the effectiveness of consistency regularization. This proposal is essential for consistency regularization as it filters out noisy samples. In appendix A.4, we show that our proposed confidence selection can benefit existing test-time optimization methods.
>
> > 2. "I'm not sure if the four ImageNet variants are really OOD data? What about other tasks?"
>
> We follow the same OOD evaluation as in the original paper of CLIP ([Radford et al., 2021](https://arxiv.org/pdf/2103.00020.pdf)), where the four ImageNet variants are referred to as "natural distribution shifts". Other works have also adopted the same datasets for OOD evaluation, e.g. [Mao et al., 2021](https://arxiv.org/pdf/2105.07926.pdf), [Kumar et al., 2022](https://arxiv.org/pdf/2202.10054.pdf).
> Nonetheless, we agree that including a more diverse set of OOD data can further demonstrate the effectiveness of TPT.
> We plan to conduct an additional evaluation on [ROBINv1 benchmarks](https://www.ood-cv.org/index.html), which covers 6 OOD scenarios including shape, 3D pose, texture, context, weather, and occlusion. We will include this evaluation in our final version.

---

> > ### Comment · Reviewer_Tt1G · 2022-08-02
> > **Reply to Author Response**
> >
> > I've carefully read the authors' responses to other reviewers and me. They have added sufficient additional experiments to support their claims and address the concerns raised by reviewers. I'd like to keep my original recommendation.

---

### Official Review · Reviewer_SAYF · 2022-07-11

**Rating:** 5
**Confidence:** 4
**Soundness:** 4 excellent
**Presentation:** 3 good
**Contribution:** 3 good

**Summary:**

The method proposes entropy-based optimization of prompt tokens during test time. To this end, the image is augmented with different views to make a batch of 64 samples. Noisy samples are filtered based on the entropy score. The Adam-based optimizer steers the tokens' representation with one step.
An ensemble with a previous baseline (e.g., CoCoOp) significantly improves various classification tasks.




**Questions:**


Can the authors provide results of ensemble between different methods?
What classes are improved by applying TPT?


**Limitations:**

yes

**Strengths And Weaknesses:**

Strengths: The work proposes a new approach (TPT) for zero-shot image classification that does not require pre-training and work during inference. The method is plausible and intuitive and suggests some novel elements, such as the confidence selection term. One benefit of this approach compared to CoCoOp is that it doesn't require additional training. In most experiments that check classification capabilities, an ensemble of TPT with  CoCoOp shows a significant improvement in several presents compared to the CoCoOp baseline.

Weaknesses: [W1] The method alone is on par with previous works (i.e., approximately 1% difference between a naive "hand-crafted" ensemble). The most significant gap comes from the ensemble of CoCoOp with TPT. However, as an ensemble, in general, is known to improve performance, the method should check other ensembles, e.g., CoCoOp (with different seeds), CoCoOp + CoOp, CoCoOp + Ensemble,  etc.
[W2] The work lacks discussions and qualitative study. The study does not contribute much to the reader in its current form. Yes, metrics are improved. But why? Which classes improved? I suggest the authors follow this work in the analysis [1].
[W3] I'm not sure the method is significantly new. I would appreciate a better discussion on the differences from MEMO.

Typos:  Tab. 1 caption requires -> require; L293 is -> are.


Refs:
[1] - The effects of regularization and data augmentation are class dependent; Balestriero et al.

---

> ### Author Response · Authors · 2022-08-02
> **Reply to Reviewer SAYF (part 1)**
>
> Thanks for your valuable feedback and suggestions. We have fixed the mentioned typos in our revision. We conducted new experiments per your suggestion and would like to address your concerns below.
>
> > Q1: The method alone is on par with previous works (i.e., approximately 1% difference between a naive "hand-crafted" ensemble). The most significant gap comes from the ensemble of CoCoOp with TPT. However, as an ensemble, in general, is known to improve performance, the method should check other ensembles, e.g., CoCoOp (with different seeds), CoCoOp + CoOp, CoCoOp + Ensemble, etc.
>
> Thanks for the suggestion. We include results of other baselines as follows, as well as in our revision in Appendix A.3. We could observe that our method (TPT + CoOp) still achieves the best performance with CLIP-RN50.
>
> |                            |  ImageNet | ImageNet-A | ImageNet-V2 | ImageNet-R | ImageNet-Sketch | Average   | OOD Average |
> |----------------------------|:---------:|:----------:|:-----------:|:------------:|:-----------------:|:-----------:|:-------------:|
> | CLIP-RN50                  | 58.16     | 21.83      | 51.41       | 56.15      | 33.37           | 44.18     | 40.69       |
> | hand-crafted ensemble      | 59.81     | 23.24      | 52.91       | **60.72**      | 35.48           | 46.43     | 43.09       |
> | CoOp                       |     63.33 |      23.06 |       55.40 |      56.60 |           34.67 |     46.61 |       42.43 |
> | CoOp ensemble (3 different seeds)   |     61.66 |      22.96 |       54.14 |      57.89 |           34.94 |     46.32 |       42.48 |
> | CoOp + hand-crafted ensemble   |     63.60 |      23.23 |       55.63 |      57.07 |           34.84 |     46.87 |       42.69 |
> | CoCoOp                     |     62.81 |      23.32 |       55.72 |      57.74 |           34.48 |     46.81 |       42.82 |
> | CoCoOp ensemble (3 different seeds) |     63.34 |      24.27 |       56.12 |      58.24 |           35.46 |     47.49 |       43.52 |
> | CoCoOp + hand-crafted ensemble |     63.03 |      24.16 |       55.73 |      57.88 |     35.22 |     47.20 |       43.25 |
> | CoCoOp + CoOp              |     63.86 |      23.69 |       56.45 |       57.7 |            35.5 |     47.44 |       43.34 |
> | TPT (Ours)                 | 60.74     | 26.67      |   54.70     |  59.11     |    35.09        |   47.26   |   43.89     |
> | TPT (Ours) + CoOp          | **64.73** | **30.32**  | **57.83**   |   58.99    | **35.86**       | **49.55** | **45.75**   |
> | TPT (Ours) + CoCoOp        | 62.93     | 27.40      |   56.60     |  59.88 |    35.43        |   48.55   |   44.83     |
>
> Additionally, we emphasize that there exists a fundamental difference between our "ensemble" (e.g., TPT + CoOp) and conventional model ensemble (e.g.,  naive "hand-crafted" ensemble, CoCoOp with different seeds). As our method (TPT) works solely at test time and uses a pre-defined prompt (which may come from a hand-crafted prompt, CoOp, or CoCoOp) as the initialization, it is complementary to hand-crafted prompts, CoOp and CoCoOp.
>
> Specifically,  taking "TPT + CoCoOp" as an example, it is done in the following steps:
> Step (1). Using the prompt output by CoCoOp as the prompt initialization;
> Step (2). Running TPT to tune this prompt at test time to get the final result.
>
> However, conventional model ensemble methods (e.g., the naive "hand-crafted" ensemble, CoOp + CoCoOp, etc.) aggregate the predictions obtained from different prompts (e.g. CoOp's prompt and CoCoOp's prompt for CoOp + CoCoOp).
>
> In addition, we would like to point out that the significant improvement brought by the naive “hand-crafted ensemble” mainly comes from ImageNet-R. However, all the rendition types (e.g., video game, origami, sculpture) in ImageNet-R are presented in the hand-crafted prompts. For example, one of the 80 prompts in the ensemble is "a sculpture of a _". This could be regarded as leakage of the domain information of the test data.
>
>
>
> (We address the rest of your concerns in the next reply due to the length limitation.)

---

> ### Author Response · Authors · 2022-08-02
> **Reply to Reviewer SAYF (part 2)**
>
> > Q2:  The work lacks discussions and qualitative study. The study does not contribute much to the reader in its current form. Yes, metrics are improved. But why? Which classes improved? I suggest the authors follow this work in the analysis [1].
>
> We thank the reviewer for the helpful suggestion. We analyze the performance of TPT among different classes.
> Specifically, we compare the per-class top-1 accuracy of a CLIP model on ImageNet-A (200 classes in total), with and without TPT. From [the plot](https://ibb.co/q1FpRc5), we find that TPT can improve model performance in most classes.
>
> In addition, we include more qualitative studies, where we demonstrate the effect of TPT on the prediction distribution of the augmented images. As shown in [this example](https://ibb.co/F45190B), we can see that TPT can effectively make the predictions more consistent across different augmented views (bottom figure). After performing TPT, the model could provide the corrected prediction (top-right figure).  More examples can be found in our revision in Appendix A.8.
>
> We are also conducting similar case studies as in [1] to analyze the effect of TPT with different hyper-parameters on each individual class. The case studies are being finalized and will be included in our final version.
>
> Lastly, we would like to mention that the main goal of our work is very different from [1], as [1] is dedicated to studying the class-dependent property of regularizations and data augmentations. However, our goal is to boost the zero-shot generalization of foundation models. Nevertheless, we thank the reviewer for referring us to [1] and suggesting a new perspective for our ablation study.
>
> > Q3. I'm not sure the method is significantly new. I would appreciate a better discussion on the differences from MEMO.
>
> Thanks for pointing it out. We discussed the differences in the last paragraph of related work, and we have updated the discussion with a more detailed comparison (as below) in our revision in Section 2.
> One of the challenges of test-time optimization is to choose a group of parameters for optimization. It is one of the major differences between our method and previous test-time optimization methods (e.g., TENT and MEMO). TENT updates the Batch-norm statistics, and MEMO optimizes the entire model. Our method, TPT, on the other hand, optimizes the text prompt while keeping the model intact. Our motivation is to avoid distorting the pre-trained features so that TPT preserves the zero-shot generalization ability of pre-trained models. As shown in our ablation study in Section 5, directly optimizing model parameters (as MEMO) does not achieve comparable improvement as our prompt-based optimization.
>
> In addition, in terms of the optimization algorithm, we refine the entropy minimization by proposing "**confidence selection**". It is designed to filter out the noisy examples before doing entropy minimization. It is an important step for test-time optimization since noisy examples may mislead the model and hurt the final performance. Such refinement is neglected in current test-time optimization literature. In appendix A.4, we show that the proposed confidence selection can be applied to other test-time optimization methods such as MEMO to further improve their performance.

---

> > ### Comment · Reviewer_SAYF · 2022-08-06
> > **Thank you for the comment. I have further questions regarding improvements.**
> >
> > Thank you for taking the time to comment. I am satisfied with some of the answers provided. As additional clarification, I would appreciate it if you could clarify the first figure you shared about improvements per class. The figure does not include class names.
> >
> > Is it possible to share which classes your method helped the most and your opinion for a reason? And when the method did not improve performance and may even have decreased it. Do you have an explanation for why it did not result in a positive outcome?

---

> > > ### Author Response · Authors · 2022-08-07
> > > **To answer your question, we update the figure and provide more analysis.**
> > >
> > > Thanks for the feedback. We have updated the figure with annotations of classes. In [the updated figure](https://ibb.co/cb9mgbT), we mark the 5 classes with the most improvement with green labels, and the 5 classes with the least improvement are in red. (Labels in bold are the Top-1 most and least improved classes.)
> > >
> > > From the labels, we do not observe a clear pattern (e.g., whether or not they are all animals/plants/sports/...; whether or not they are all small/large objects) among the most or the least improved classes. To further investigate the per-class performance, we conduct the same analysis on the ImageNet-R dataset (with 200 classes in total). By looking at the [analysis on ImageNet-R](https://ibb.co/rdz8zLp), we do not find an overlap between the most improved classes of the two datasets, nor between the least improved classes. From this observation, we presume that TPT's behavior is not associated with the semantics of classes.
> > >
> > > As we do not observe a class-specific behavior of our method, we would like to explain why TPT works from the perspective of consistency regularization. We leverage an assumption that a robust model's decision boundaries should lie in low-density data regions ([Chapelle & Zien, 2005](http://proceedings.mlr.press/r5/chapelle05b/chapelle05b.pdf)). Consistency-regularization-based methods (e.g., SimCLR ([Chen et al, 2020](https://arxiv.org/pdf/2002.05709.pdf)), BYOL ([Grill et al., 2020](https://arxiv.org/pdf/2006.07733.pdf))) achieve this goal by making the network's output invariant to small input perturbations (e.g., augmentations). In our work, we use consistency regularization as our test-time prompt tuning objective with the confidence selection module. We assume that an optimal prompt should lead to robust feature representation regardless of different data augmentation views.
> > >
> > > We further include individual analysis on samples for which TPT does not result in a positive outcome. We take the class "canoe" from ImageNet-A (the least improved class) as an example. We randomly choose an image that is classified incorrectly after applying TPT.
> > > For [this image](https://ibb.co/JsfWFhX), the ground-truth is "canoe" but it is misclassified as "chihuahua" after applying TPT. Although incorrect, we think it is reasonable for CLIP to make this prediction after the prompt is tuned. Note that CLIP classifies images by matching the image with text descriptions, which consist of the combination of a text prompt (e.g. "a photo of a __ " ) and all possible classes. Even for the same image, CLIP's prediction may change given different prompts (i.e., context ([Zhou et al., 2021](https://arxiv.org/pdf/2109.01134.pdf))). Thus, when provided with a text prompt that emphasizes the middle of the image, it is reasonable for CLIP to predict the image as the center object (the "chihuahua" in the given example). As TPT encourages consistency among different augmented views, it is possible that the prompt is optimized in favor of the center object. We will include more case studies in our final version. Thanks for the suggestion.

---

> > > > ### Author Response · Authors · 2022-08-08
> > > > **Kindly look forward to further discussions**
> > > >
> > > > We appreciate your time and constructive comments. As we are approaching the end of the discussion period, we would like to hear your feedback on our response to your newly raised questions. We would appreciate the opportunity to engage further if needed.
> > > >
> > > > In our response above, we provide further information regarding your questions. We update the annotations in our figures of per class accuracies. We also conduct the per-class analysis on ImageNet-R. In addition, we include a case study to provide further insight.
> > > >
> > > > We sincerely hope to hear your feedback.

---

> > > > > ### Comment · Reviewer_SAYF · 2022-08-09
> > > > > **Many thanks for your response**
> > > > >
> > > > > Thanks for the elaborated analysis. I agree with the conclusion that the drop might also be attributed to data issues, which I hope you can explore in more detail. An explanation for improvements might also be found in a qualitative study. Maybe improved classes usually have a center object, as it seems not related to semantics.
> > > > >
> > > > > My main concern is why the augmented optimization must occur during inference time, which is computationally expensive, rather than using a more CoOp-like procedure.  It may also be beneficial to add additional augmented views to CoOp. It could have been claimed that the authors' method requires no pre-training at all. Nevertheless, the final approach uses an ensemble with CoOp; therefore, it also uses pre-trained CoOp.
> > > > >
> > > > > However. I'm leaning towards acceptance since the method improves classification robustness and suggests a new way to optimize classification tasks in a zero-shot manner.
> > > > >
> > > > > To authors, consider discussing other test-time optimization techniques to steer large model predictions, e.g., [1] for language generation and [2] for captioning.
> > > > > [1] Plug and Play Language Models: A Simple Approach to Controlled Text Generation; ICLR'20
> > > > > [2] ZeroCap: Zero-Shot Image-to-Text Generation for Visual-Semantic Arithmetic; CVPR'22

---

> > > > > > ### Author Response · Authors · 2022-08-09
> > > > > > **Thanks for your reply and recommendation.**
> > > > > >
> > > > > > Thanks for your reply and recommendation. We will include a discussion of [1, 2] in our related work section. We also plan to further investigate the per-class performance on both dropped and improved classes. We will include additional analysis and qualitative study in our final version.
> > > > > >
> > > > > > In response to your concern, we consider our main contribution to be proposing a test-time prompt tuning method for pre-trained vision-language models, which works on a single test sample. The pre-trained model could be equipped with either a hand-crafted prompt (e.g., CLIP) or learned prompts (e.g., CoOp, CoCoOp). Therefore, we mainly focus on the comparison of CLIP vs. CLIP+TPT, CoOp vs. TPT+CoOp, etc. We show that TPT can consistently improve the generalization of these models.
> > > > > >
> > > > > > We agree that the augmented optimization could occur at other stages (e.g., the training stage). In this work, we focus on improving the model generalization at the test stage so that we can adapt to test samples from unseen distributions. Thanks for the constructive suggestion. We plan to explore it in the future as future work.

---

> > > > ### Author Response · Authors · 2022-08-09
> > > > **Sincerely look forward to your feedback**
> > > >
> > > > We sincerely hope to get your feedback on our most recent response and see if we have solved your concerns.
> > > > We would appreciate the opportunity to provide further clarification if needed. We would also appreciate it if you consider raising your rating of this submission given our additional results and analysis.

---

### Meta-Review · Area_Chair_PXoN · 2022-08-23

**Recommendation:** Accept
**Confidence:** Certain

**Metareview:**

This paper proposes a technique for training prompts for open-vocabulary vision models (e.g. CLIP) at test time, i.e. without any labeled data. The model is trained to minimize the entropy of the average prediction of many augmented views of a test image. This improves performance to varying degrees without requiring any additional labeling. The method and approach are interesting and some useful analysis is provided. Reviewers agreed that the paper should be accepted. Beyond the suggested changes made by reviewers, I'd recommend some additional references to better situate the paper with respect to past work on consistency regularization, entropy minimization, and transductive learning.

**Award:**

No

---

### Decision · Program_Chairs · 2022-09-14

Accept